# Plasmalogen Improves Memory Function by Regulating Neurogenesis in a Mouse Model of Alzheimer’s Diseases

**DOI:** 10.3390/ijms241512234

**Published:** 2023-07-31

**Authors:** Rongzi Li, Wei Xiong, Boying Li, Yixuan Li, Bing Fang, Xifan Wang, Fazheng Ren

**Affiliations:** 1Key Laboratory of Precision Nutrition and Food Quality, Department of Nutrition and Health, China Agricultural University, Beijing 100083, China; rzl0903@126.com (R.L.); xiongwei910702@126.com (W.X.); lboying@126.com (B.L.); liyixuan@cau.edu.cn (Y.L.); bingfang@cau.edu.cn (B.F.); 2Food Laboratory of Zhongyuan, Luohe 462000, China

**Keywords:** Alzheimer’s disease, plasmalogen, memory, neurogenesis, neural stem cells

## Abstract

Adult hippocampal neurogenesis (AHN) is associated with hippocampus-dependent cognitive function, and its initiation is attributed to neural stem cells (NSCs). Dysregulated AHN has been identified in Alzheimer’s disease (AD) and may underlie impaired cognitive function in AD. Modulating the function of NSCs and stimulating AHN are potential ways to manipulate AD. Plasmalogen (PLA) are a class of cell membrane glycerophospholipids which exhibit neuroprotective properties. However, the effect of PLA on altered AHN in AD has not been investigated. In our study, PLA(10μg/mL) -attenuated Aβ (1-42) (5μM) induced a decrease in NSC viability and neuronal differentiation of NSCs, partially through regulating the Wnt/β-catenin pathway. Additionally, AD mice were supplemented with PLA (67mg/kg/day) for 6 weeks. PLA treatment improved the impaired AHN in AD mice, including increasing the number of neural stem cells (NSCs) and newly generated neurons. The memory function of AD mice was also enhanced after PLA administration. Therefore, it was summarized that PLA could regulate NSC differentiation by activating the Wnt/β-catenin pathway and ameliorate AD-related memory impairment through up-regulating AHN.

## 1. Introduction

Adult hippocampal neurogenesis (AHN) is a process in which neural stem cells (NSCs) in the adult brain hippocampus continue to proliferate and differentiate to produce new neurons [1]. Evidence suggests that AHN is crucial for hippocampus-dependent processes, including learning and memory [2,3]. Dysregulated neurogenesis has been identified in various brain disorders, including Alzheimer’s disease (AD) [4], which is the most prevalent form of dementia characterized by deterioration of cognitive functions and progressive memory loss. AD patients experience a wide range of symptoms as the disease progresses, including difficulty thinking, memory lapses, and mood swings, resulting in reduced self-care capacity [5]. The hippocampus is among the first regions to incur damage in a brain affected by AD. The altered AHN in early-stage life is associated with severe neuronal vulnerability in AD later in life, resulting in increased neuronal death and cognitive decline [6]. Improving AHN was shown to ameliorate AD pathology and improve cognitive function in AD [6]. Therefore, AHN enhancers would be a potentially powerful treatment strategy for AD. 

Multiple signaling pathways tightly regulate AHN [7]. There is growing evidence that Wnt signaling contributes to controlling various phases of neurogenesis, including the activation of quiescent NSCs, proliferation, and the generation of new neurons [8,9]. Wnt ligands could affix to the Wnt receptors on NSCs and trigger the activation of the Wnt/β-catenin pathway [9]. The downregulation of Wnt signaling was observed in AD patients and animals, and this may be related to the impaired neurogenesis in AD and the pathophysiology of AD [10,11]. In contrast, overexpression of Wnt3a restored AHN in an AD mouse model [6]. Pharmacological interventions, such as Lithium, Valproic acid, and Curcumin, were suggested to promote neurogenesis and cognitive performance in AD animals by regulating Wnt signaling [12,13,14]. 

Plasmalogen (PLA) are a class of cell membrane glycerophospholipids that have a vinyl-ether bond at the sn-1 position and an ester bond at the sn-2 location. Emerging evidence suggests a significant correlation between PLA and AD, as a pilot study revealed a 70% reduction of PLA in AD patients’ brains compared to healthy individuals [15]. PLA functions as a scavenger to guard other lipids against oxidative damage because of its structure. Decreased PLA promotes further ongoing oxidative damage in AD [16]. On the other hand, PLA are major components of cell membranes and are associated with normal synaptic structure and function and neurotransmitter release. Synaptic dysfunction and neurotransmitter deficits in AD may be exacerbated by PLA loss [17]. A recent study showed that plasmalogen intervention could enhance NSCs proliferation in the hippocampus of aged mice [18]. However, the effect of PLA on altered neurogenesis in AD has yet to be investigated. In this study, we treated NSCs with amyloid-β (1-42) (Aβ (1-42)), detecting the cell viability and neuronal differentiation of NSCs. Additionally, we supplemented AD mice with PLA for 6 weeks, and assessed the effect of PLA on AHN and the hippocampal-dependent cognitive function of AD mice.

## 2. Results

### 2.1. PLA-Attenuated Aβ (1-42) Induced Decrease in NSC Viability

The C17.2 cells were subjected to different concentrations (0 μM, 1 μM, 2.5 μM, 5 μM, 10 μM, 20 μM) of Aβ (1-42) for 24 h. The Aβ (1-42) treatment led to a dose-dependent reduction in the viability of C17.2 cells (Figure 1a). Notably, the viability of cells was significantly reduced beyond a concentration of 2.5 μM of Aβ (1-42). Microscopic analysis revealed substantial cell death upon administration of higher doses of Aβ (1-42). Therefore, we employed a concentration of 5 μM Aβ (1-42) in subsequent studies, which decreased the viability of C17.2 cells to 63.3% (*p* < 0.01) of the control (Figure 1a). Next, cells were pre-treated with PLA with various concentrations (1 μg/mL, 5 μg/mL, 10 μg/mL, 20 μg/mL) for 2 h and subsequently with Aβ (1-42) for 24 h (Figure 1b). The vitality of Aβ (1-42)-treated NSCs was significantly enhanced after upwards of 5 μg/mL PLA administration, compared to cells solely treated with Aβ (1-42). As a result, our findings imply that PLA may possess the ability to mitigate the deleterious impact of Aβ (1-42) on the cell viability of NSCs. 10 μg/mL PLA was used in subsequent studies. 

### 2.2. PLA-Attenuated Aβ (1-42) Induced Decrease in Neuronal Differentiation of NSCs

To examine the effect of Aβ (1-42) and PLA on the differentiation of NSCs, C17.2 cells were either administrated with Aβ (1-42) (5 μM) or Aβ (1-42) (5 μM) and PLA (10 μg/mL) in a differentiation medium for 7 days. Following this, we extracted total RNA from cells subjected to different treatments and measured the mRNA level of *Nestin* and *MAP2*, which serve as NSC and mature neuronal markers, respectively. Our results indicated a down-regulation of *MAP2* and an up-regulation of the mRNA level of *Nestin* in Aβ (1-42)-stimulated C17.2 cells compared with the control. However, administration of PLA reversed the changes induced by Aβ (1-42) (Figure 2a,b). Subsequently, we stained the cells with Nestin and MAP2 antibodies to better distinguish NSCs and mature neuron. As shown in Figure 2c, a lower density of neurites and a lower length of neurites were observed in Aβ (1-42)-stimulated C17.2 cells in comparison to untreated cells. However, the length of neurites was significantly increased after PLA treatment. These results were presented by MAP2 staining. In contrast, the number of Nestin positive NSCs was lower in the Control group and Aβ (1-42) +PLA group, compared to Aβ (1-42) group (Figure 2d). Overall, these results confirmed that PLA-attenuated Aβ (1-42) induced a decrease in the neuronal differentiation of NSCs. 

### 2.3. PLA Improved Neuronal Differentiation of NSCs by Activating the Wnt/β-Catenin Pathway

Considering the essential role of the Wnt/β-catenin pathway in regulating the differentiation of NSCs, we sought to investigate whether the impact of PLA on Aβ (1-42)-induced changes was involved in the modulation of this pathway. Following a 7-day differentiation period, cell proteins were collected and the expression of Wnt3a, pGSK3β, and β-catenin was evaluated. In comparison to the Control group, the Aβ (1-42)-stimulated group had lower levels of Wnt3a, pGSK3β/GSK3β ratio, and β-catenin. However, the level of those proteins was significantly enhanced after PLA supplementation (Figure 3). Our findings imply that PLA could reverse the Aβ (1-42)-induced decrease in the neuronal differentiation of NSCs, at least in part by activating the Wnt/β-catenin pathway. 

### 2.4. PLA Improves Hippocampal Neurogenesis in AD Mice

AHN decreases in the early phases of AD. To confirm the effect of PLA on neurogenesis in vivo, we orally administered PLA to APP/PS1 mice for 6 weeks. We assessed AHN in our animals by evaluating the number of new generated neurons that were DCX antibody labelled. The staining results revealed that, when compared to the Control group, an 81% (*p* < 0.001) reduction of DCX-expressing cells was seen in the DG of the hippocampus of the AD group mice. PLA treatment, on the other hand, significantly increased the quantity of DCX-expressing cells in AD animals (Figure 4). According to these findings, PLA promotes AHN in AD mice.

Next, the number of hippocampal SOX2+ cells were determined to assess the impact of PLA on NSC populations. A 24.7% (*p* < 0.001) smaller number of SOX2+ cells were seen throughout the hippocampus of AD mice. Notably, administration of AD mice with PLA restored decreased NSC populations in mice hippocampus, leading to a significant improvement (Figure 5). These outcomes suggest that PLA could protect against AD-induced decreased NSCs in mice hippocampus. 

### 2.5. PLA Improves Hippocampal-Dependent Memory Function in AD Mice

The elevation of AHN has been associated with memory function [4]. We also detected whether PLA supplementation improves hippocampal-dependent memory impairment in AD. NOR task and Y-maze were conducted to evaluate mice recognition memory and spatial memory, respectively. During the NOR task (Figure 6a), following exposure to two identical objects (familiarization trial), recognition memory was assessed by substituting one of the familiar ones for a new object in the subsequent trials. During the familiarization trial, mice from various groups explored the two objects for the same times. However, although there was no significant difference in the total exploration time in the choice trial among groups (Figure 6b), the Control group mice clearly preferred to investigate the novel one, whereas the AD group mice had a similar preference for different objects. The decreased discrimination index ratio in the AD group represented this result (Figure 6c). In contrast, after PLA supplementation, AD mice preferred novel objects in the choice trial and had a significantly increased discrimination index ratio (Figure 6c), suggesting that PLA suppressed recognition memory deterioration in AD mice. 

Next, we used the Y-maze test (Figure 7a) to evaluate the spontaneous alternation ratio, an indicator of the spatial memory function of animals. As shown in Figure 7b, no significant difference in total exploration times was observed among groups. In contrast, AD mice exhibited a significant deficit in spontaneous alternation in comparison to the Control group mice, indicating the impaired spatial memory function in AD mice. However, significant restoration of spontaneous alternation was observed after treatment with PLA (Figure 7c). Hence, similar to recognition memory, AD mice performed worse in spatial memory tests, whereas PLA administration effectively protected against this deterioration in AD mice. 

## 3. Discussion

Impaired neurogenesis has been related to synaptic and cognitive dysfunction during the progression of AD [19,20]. Increasing neurogenesis in the early stages of AD could delay the death of neuronal cells and serve as a unique therapeutic approach to the disease. Therefore, finding neurogenesis stimulants in AD is a crucial question. Our study hypothesized that PLA is a neurogenic stimulant, based on its antioxidant and anti-inflammatory properties and its close relationship with synaptic function. Our findings suggest that PLA-attenuated Aβ (1-42) induced a decrease in the neuronal differentiation of NSCs by activating the Wnt/β-catenin pathway. Moreover, in vivo study indicates that PLA could improve hippocampal neurogenesis and suppress hippocampal-dependent behavioral deficits in AD mice.

The process of AHN starts with NSCs, which could self-renew or differentiate into neuroblasts. A growing body of research has been conducted to identify drugs or treatments that can promote the survival, proliferation, and differentiation of NSCs [21,22]. Moreover, the modulation of NSCs has been linked to cognitive improvement [23]. However, a gradual decline in NSC population and compromised differentiation ability occurs in certain neurodegenerative diseases. In AD, Aβ peptides and oligomers exert a detrimental effect on NSCs, leading to cell death and impaired function [24]. Aβ (1-42) administration induced cytotoxicity in NSCs and alterations in AHN in mouse brains, as reported in previous studies [25,26]. To explore the potential of PLA in mitigating Aβ (1-42)-induced cytotoxicity in NSCs, we subjected C17.2 NSCs to Aβ (1-42) treatment and evaluated their cell viability and differentiation capacity. Our results showed a decline in cell viability and neuronal differentiation of NSCs following Aβ (1-42) exposure. Nevertheless, the administration of PLA effectively reversed these adverse effects.

Signals provided by the neurogenic niche modulate the process of AHN. It was proposed that Wnt signals play an essential role in regulating neurogenesis [9]. The combination of Wnt ligands and the Wnt receptor leads to the inhibition of GSK3β, which is phosphorylated, finally raising the level of cytosolic β-catenin. Subsequently, accumulated β-catenin moves to the nucleus and promotes the transcription of Wnt target genes. These genes have been implicated in different stages of AHN [27]. In our study, after NCS differentiation, cell proteins were extracted and analyzed. Decreased Wnt3a, pGSK3β/GSK3β, and β-catenin levels were observed in the Aβ (1-42)-treated group in comparison to the control group. However, co-treatment PLA with Aβ (1-42) suppressed the reduction in Wnt3a, pGSK3β/GSK3β and β-catenin. Therefore, the Wnt/β-catenin pathway may be involved in the modulation of PLA on Aβ (1-42)-induced NSC damage.

Subsequently, we supplemented AD mice with PLA and examined the change in AHN. Firstly, we detected the number of newborn neurons using the DCX antibody. Comparing AD mice to wide-type control mice, a considerably smaller proportion of DCX-expressing cells were found in the hippocampus. Moreover, the NSC number was also reduced in the AD mice. However, the administration of PLA reversed the decrease in both the number of newborn neurons and NSCs in AD mice. Hence, our findings indicate that PLA could ameliorate AHN disruption in AD animals. 

AHN has been implicated in hippocampal-dependent cognitive function. It was reported to chemically or genetically impair AHN, including the ablations of NSCs and the inhibition of proliferation, causing hippocampus-dependent learning and memory to be compromised [19,28]. However, specifically improving newborn neurons’ survival in the DG was shown to enhance cognitive function associated with the hippocampus [29]. Therefore, the connection between AHN and hippocampal-dependent cognitive function has been established, so targeting AHN will be beneficial for cognition and this calls for additional research. This study also examined hippocampal-dependent behavioral changes in AD mice with or without PLA supplementation. NOR tests and Y-maze tests results indicate that PLA restored decreased recognition and spatial memory in AD mice. 

A few studies have also reported the neuroprotective effects of PLA on AD. For example, PLA could regulate oxidative stress and inflammatory response in AD due to its antioxidant and anti-inflammation effects. The loss of PLA may play a role in AD-related synaptic dysfunction since PLAs are crucial lipids that facilitate the membrane fusion of synaptic vesicles linked to neurotransmitter release [16,30]. Furthermore, PLA administration has been shown to inhibit Aβ production by regulating γ–secretase activity and to prevent neuronal cell death due to its anti-apoptotic action [31,32]. In summary, based on previous evidence and our study, PLA supplementation may be an effective strategy to address the pathophysiology of AD. 

Neuroprotective effects of PLA may be partially due to the crucial fatty acids carried at the sn-2 position of PLA. These fatty acids are mainly docosahexaenoic acids (DHA) and eicosapentaenoic acid (EPA) [33]. Phospholipase A2 (PLA2), which cleaves the acyl chain at the sn-2 position, can release free fatty acid DHA and EPA from PLA [33]. The association between DHA/EPA and AD has been established in previous studies. For example, either DHA or EPA could inhibit β-secretase and enhance α-secretase activity, promoting the non-amyloidogenic processing rather than the amyloidogenic processing of amyloid precursor protein [34,35]. In addition, EPA or DHA could improve Aβ degradation and clearance, finally leading to a decreased level of Aβ and amyloid fibril formation [36,37]. 

## 4. Materials and Methods

### 4.1. Plasmalogen Preparation

300 g scallop was homogenized by a blender in *n*-hexane/isopropanol (3:2, *v*/*v*). The mashed sample was transferred to a bottle and shaken thoroughly at high speed for 30 min before centrifuging at 1000 *g* for 5 min at room temperature. Then, we added saturated anhydrous sodium sulfate solution to the supernatant and transferred the liquid to a separatory funnel. After 60 min, the upper layer of liquid in the separation funnel was transferred to a round-bottomed flask which connects to a rotary evaporator. Subsequently, the yielded total lipid was dissolved in ice-cold *n*-hexane/acetone (1:1, *v*/*v*) and then went through an enzymatic hydrolysis process to obtain PLA according to previous study [33]. Ultra-high performance liquid chromatography with quadrupole time-of-flight mass spectrometry (UPLC-Q-TOF) were used to examine and verify the extracted PLA (Table 1). The UPLC-Q-TOF system included an Agilent 1290 II UPLC system and an Agilent Technologies Q-TOF with an Agilent Jet Stream electrospray 186 source (Santa clara, CA, USA). The chromatographic separations were performed using BEH C18 (2.1 × 100 mm, 1.7 μm) UPLC column and a C18 guard column (2.1 × 10 mm, 1.7 µm) at 55 °C. The mobile phase consisted of 10 mM ammonium formate 190 in water/acetonitrile (6:4, *v*/*v*) and 10 mM ammonium formate in 191 isopropanol/acetonitrile (9:1, *v*/*v*). The Agilent Jet Stream parameters were set as follows: dry gas 300 °C, 7 L/min; sheath gas 350 °C, 12 L/min; fragmentor voltage 140 V; capillary voltage 3000 V. Data was collected by Agilent Mass Hunter Workstation Data acquisition software (Version B.10.01). The extracted PLA was freeze-dried to store or dissolve in a 2% tween solution (50 mg/mL) for the following intervention experiment.

### 4.2. Animals

Sixteen-week-old female APP/PS1 transgenic mice were obtained from Cyagen Biological Technology Co., Ltd. (Suzhou, China). Age-matched wild-type mice were purchased from the same company. Mice were kept under standard lab conditions, which included a 12-h light/dark cycle and a temperature range of 24–26 °C. Mice were divided into three groups (n = 12/group) following a week of acclimatization. The groups were as follows: the Control group (wild-type mice), the AD group (APP/PS1 mice), and the AD-PLA group (APP/PS1 mice). The AD-PLA group mice received an oral gavage of 67 mg/kg/day PLA, while the Control group and AD group mice received vehicle solution for 6 weeks (Figure 8). All experiments followed the national and international animal welfare system, approved by the Laboratory Animal Welfare and Animal Experiment Ethics Review Committee of China Agricultural University. The approval number is AW60103202-5-1 (6 January 2022)

### 4.3. Behavioral Testing

After 6 weeks of treatment, behavioral tests were conducted. Novel object recognition (NOR) tests were used to examine the recognition memory function of mice. NOR tasks were carried out according to a previous study [14]. Briefly, the test was conducted in an open-field arena. The animals were free to roam the empty arena while becoming habituated. The form tests, which include familiarization and choice trials, were carried out 24 h following habituation. During the familiarization phase, animals were given 5 min to investigate two similar objects in the arena, whereas in the choice trial, animals were free to investigate a familiar object and a novel one. There is a 1-h gap between each trial. This behavioral experiment is based on animals’ innate propensity to focus more time on unfamiliar objects than they do on familiar ones. The percent of discrimination index was determined by the following equation:Discrimination index = (exploration time with novel object)/(the total object exploration time) × 100%

The Y-maze was conducted as previously described [38] as a behavioral experiment used to assess mice spatial memory. The Y-shaped maze consists of three identical arms at a 1200 angle from each other. The mice are typically positioned at the end of one arm and given unrestricted access to all arms during the test. Animal movements are monitored. A camera counted how many mice entered each arm and how long they stayed there. Mice entering three distinct arms in succession was referred to as an alternation. The following equation could compute the spontaneous alternation ratio: Spontaneous alternation = (number of alternations)/(total arm entries − 2) × 100%

### 4.4. Tissue Collection

All the animals underwent CO_2_ anesthesia on the final day of the experiment before receiving 0.9% saline solution and 4% paraformaldehyde. The entire mouse brain was carefully detached from the skull and frozen in 4% paraformaldehyde at 4 °C for later immunofluorescence examination.

### 4.5. Brain Tissue Immunofluorescence

The fixed brains were then paraffin-embedded and followed by a 5-μm coronal section. Sections at the hippocampal level were collected and mounted onto slides, according to the Mouse Brain Atlas. The sections were dewaxed and rehydrated in xylene and graded ethanol, respectively. An antigen retrieval procedure utilizing a citrate solution was then carried out, followed by incubation in 5% BSA for 30 min at room temperature (RT). Next, brain sections were reacted with primary antibodies for an overnight period at 4 °C. Then, secondary antibodies were exposed to sections for 1 h at RT. Finally, the sections reacted with DAPI and were covered with coverslips. The primary antibodies included rabbit anti-DCX antibody (1:100, A0149) and rabbit anti-SOX2 (1:100, 11064-1-AP), which were obtained from ABclonal (Wuhan, China) and Proteintech (Wuhan, China), respectively. The secondary antibodies Goat anti-rabbit IgG, Alexa Fluor 647 (1:500, A0468) were obtained from Beyotime (Shanghai, China). The stained cells were photographed with an Airyscan LSM900 confocal microscope (ZEISS, Oberkochen, Germany). Three visual fields from each dentate gyrus (DG) zone of the hippocampus were selectejd for imaging. We recognized NSCs in the DG zone as SOX2+ nucleus cells. In the subgranular layer of the DG, DCX+ cells were identified as newborn neurons.

### 4.6. C17.2 Cell Culture 

C17.2 cells are multipotent NSCs derived from mouse cerebellum. They could differentiate into neuron or glial cells when supplemented with brain-derived neurotrophic factor (BDNF, Alomone labs# B-250, Jerusalem, Israel) and nerve growth factor (NGF, Sigma#N0513, St. Louis, MI, USA) [21]. The cells were obtained from Shanghai Cell Research Center (Shanghai, China). For routine culture, the cells were grown in a proliferation medium containing DMEM (Gibco#41965039, Grand Island, NE, USA) supplemented with 10% fetal bovine serum (Gibco#12664025C, Grand Island, NE, USA) and 100 U/mL penicillin/streptomycin (Beyotime# C0222, Shanghai, China). 

The cells were initially seeded in a proliferation medium for the differentiation test, then switched to a differentiation media 24 h later. DMEM, 100 U/mL penicillin/streptomycin, 10 ng/mL BDNF, and 10 ng/mL NGF made up the differentiation medium. To find a proper Aβ (1-42) (abcam#ab120301, Cambridge, UK) concentration for the following study, C17.2 cells were given various concentrations of Aβ (1-42) (0 μM, 1 μM, 2.5 μM, 5 μM, 10 μM, 20 μM) for 24 h. Next, the cell viability assay was conducted to examine the cell injury degree. To determine an effective concentration of PLA, the cells were exposed to 5 μM Aβ (1-42) plus various concentrations of PLA (1 μg/mL, 5 μg/mL, 10 μg/mL, 20 μg/mL). Similarly, after 24 h incubation, the cell viability assay was performed. 

### 4.7. Cell Viability Assay

In a 96-well plate, C17.2 cells (5000 cells/well) were planted. Following various treatments, cell viability was assessed using the cell-counting kit-8 test (CCK-8). Briefly, each well of the plate received a 10 ul addition of CCK-8 solution, which was then incubated for 1 h at 37 °C. A microplate reader detected absorbance at 450 nm. 

### 4.8. Cell Immunofluorescence

The cells were fixed in 4% paraformaldehyde for 10 min, then permeabilized with 0.1% Triton X-100. Next, the cells were incubated with 5% BSA for 1 h and then in primary antibodies mouse anti-Tuj1 (1:150, Beyotime#AT809, Shanghai, China) at 4 °C overnight. Then, the cells were rinsed in PBS and then incubated with Goat anti-mouse IgG, Alexa Fluor 647 (1:500, Beyotime#A0473, Shanghai, China) for 1 h at RT. Next, the cells reacted with DAPI and were covered with coverslips. The stained cells were photographed with an Airyscan LSM900 confocal microscope (ZEISS).

### 4.9. RT-PCR Analysis

Trizol reagent (Invitrogen, Carlsbad, CA, USA) was used to extract the total RNA of the cells. Next, using 5 All-In-One RT Master Mix (Applied Biological Materials#G592, Richmond, BC, Canada), RNA was reverse transcribed. Takara TB Green Fast qPCT mix (Takara#RR430, Shiga, Japan) was used in real-time PCR experiments on a 7900HT instrument (Applied Biosystems, Waltham, MA, USA) to evaluate the contents of mRNA. *GAPDH* mRNA expression level was the internal standard. RT-PCR primers sequences are listed:

*Nestin:* Fwd-5′CCCTGAAGTCGAGGAGCTG3′ Rev-5′CTGCTGCACCTCTAAGCGA3′

*MAP2*: Fwd-5′ GCCAGCCTCGGAACAAACA3′ Rev-5′GCTCAGCGAATGAGGAAGGA3′

*GFAP*: Fwd-5′ CGGAGACGCATCACCTCTG3′ Rev-5′AGGGAGTGGAGGAGTCATTCG3′

*Gapdh*: Fwd-5′TGGATTTGGACGCATTGGTC3′ Rev-5′ TTTGCACTGGTACGTGTTGAT3′

### 4.10. Western Blot

RIPA lysis buffer (Beyotime#P0013C, Shanghai, China) was used to extract NSC proteinsm and then the supernatant was obtained for Western blot analysis by centrifuging at 14,000 rpm for 15 min at 4 °C. Primary antibodies were Anti-Wnt3a antibody (1:1000, abcam#ab219412, Cambridge, UK), pGSK3β (1:1000, Beyotime#AF5830, Shanghai, China), GSK3β (1:1000, Beyotime#AG751, Shanghai, China) and β-catenin (1:1000, abcam#ab6302, Cambridge, UK). HRP-linked secondary antibodies (1:5000, Beyotime#A0218, Shanghai, China) were also utilized. The enhanced luminol-based chemiluminescent kit (34075, ThermoFisher, Waltham, MA, USA) was used to visualize the bands. GAPDH (1:5000, Affinity#AF7021, Nanjing, China) was an internal control. 

### 4.11. Statistical Analysis

The GraphPad Prism program (version 9.5.0) was used to examine all data, which were all expressed as the mean ±  SEM. At least three independent technical replicates of each experiment were run. One-way ANOVA was used to evaluate group differences, and then Turkey’s multiple comparison tests were performed. A *p*-value of 0.05 or less was deemed to be statistically significant.

## 5. Conclusions

Emerging evidence indicates that AHN impairment is a crucial pathological mechanism of AD and plays an essential role in cognitive function in AD. Currently, neurogenic stimulants have become promising therapeutic treatments for AD. Our study confirmed that PLA treatment protected NSCs from Aβ (1-42)-induced cytotoxicity by modulating the Wnt/β-catenin pathway. Further in vivo study suggested that PLA regresses AHN deterioration and improves the hippocampal-dependent cognitive function of AD mice (Figure 9). Overall, PLA is a potential candidate for a neurogenesis enhancer to slow down AD progression.

## Figures and Tables

**Figure 1 ijms-24-12234-f001:**
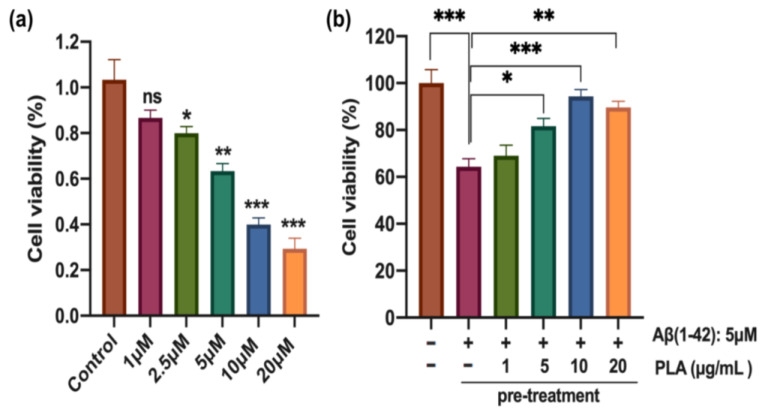
The effect of PLA on cell viability of Aβ (1-42)-treated NSCs (**a**) Aβ (1-42)-induced C17.2 cell death. (**b**) PLA-attenuated Aβ (1-42)induced decrease in NSC viability. Data was analyzed by one-way ANOVA with Tukey’s post hoc test and are shown as mean ± SEM (*n* = 3). * *p* < 0.05, ** *p* < 0.01 and *** *p* < 0.001. ns: no statistically significant difference.

**Figure 2 ijms-24-12234-f002:**
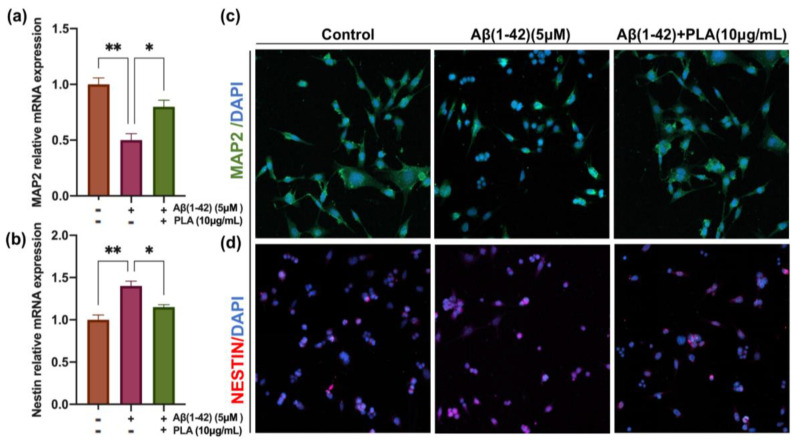
The effect of PLA (10 μg/mL) on differentiation capacity of Aβ (1-42) (5 μM)-treated NSCs (**a**,**b**) The mRNA levels of *MAP2* and *Nestin* in NSCs after 7 days differentiation. (**c**) MAP2 immunofluorescence staining. (**d**) NESTIN immunofluorescence staining. Data was analyzed by one-way ANOVA with Tukey’s post hoc test and are shown as mean ± SEM (*n* = 3). * *p* < 0.05, ** *p* < 0.01.

**Figure 3 ijms-24-12234-f003:**
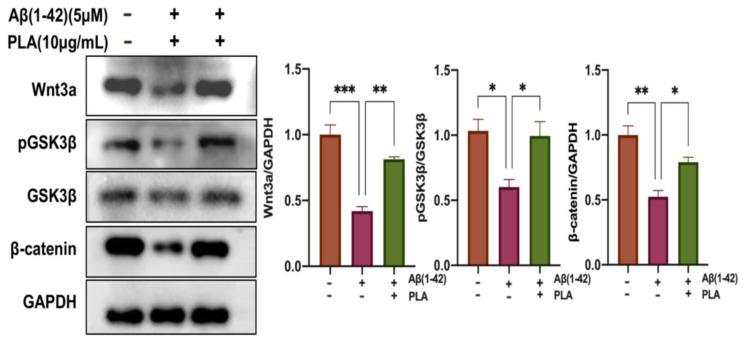
Western blot analysis of the Wnt/β-catenin pathway. Data was analyzed by one-way ANOVA with Tukey’s post hoc test and are shown as mean ± SEM (*n* = 3). * *p* < 0.05, ** *p* < 0.01 and *** *p* < 0.001.

**Figure 4 ijms-24-12234-f004:**
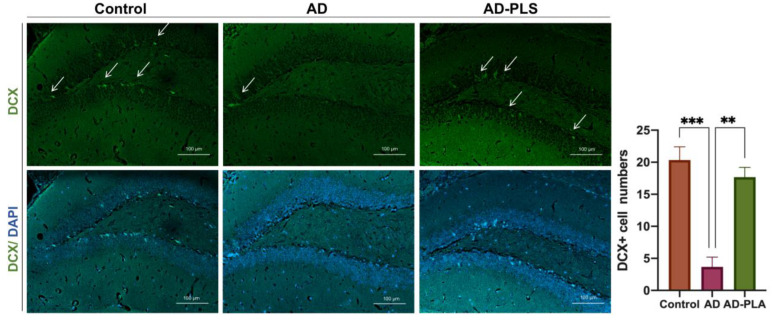
The number of newly generated neurons in the hippocampus of mice. Arrows mark the DCX-expressing cells. Data was analyzed by one-way ANOVA with Tukey’s post hoc test and are shown as mean ± SEM (*n* = 3). ** *p* < 0.01, *** *p* < 0.001.

**Figure 5 ijms-24-12234-f005:**
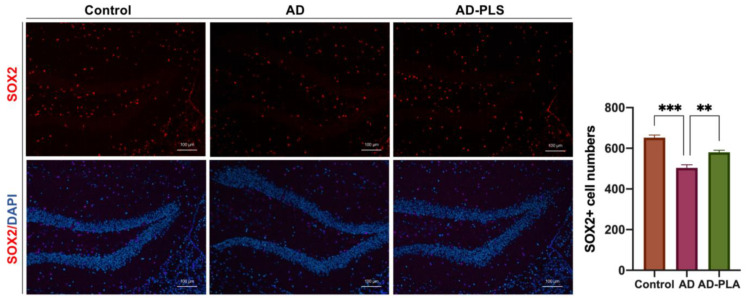
The number of neural stem cells in the hippocampus of mice. Data was analyzed by one-way ANOVA with Tukey’s post hoc test and are shown as mean ± SEM (n = 3). ** *p* < 0.01, *** *p* < 0.001.

**Figure 6 ijms-24-12234-f006:**
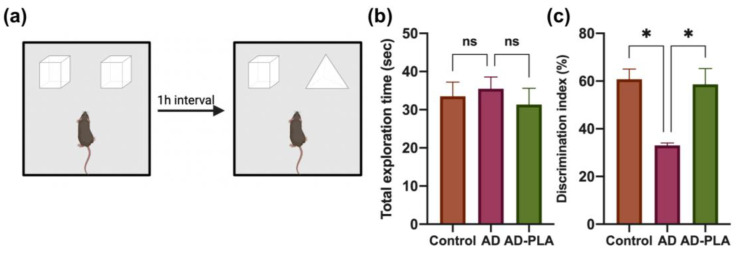
The effect of PLA on recognition memory of AD mice. (**a**) Diagram of the Novel Object Recognition test. (**b**) Total exploration time in the choice trial. (**c**) Discrimination index ratio. Data was analyzed by one-way ANOVA with Tukey’s post hoc test and are shown as mean ± SEM (*n* = 6). * *p* < 0.05. ns: no statistically significant difference.

**Figure 7 ijms-24-12234-f007:**
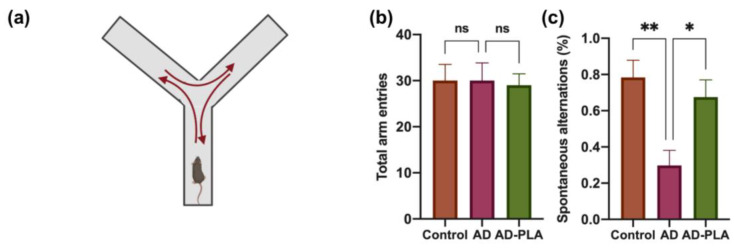
The effect of PLA on spatial memory of AD mice. (**a**) Diagram of Y-maze test. (**b**) Total arm entries. (**c**) Spontaneous alternation ratio. Data was analyzed by one-way ANOVA with Tukey’s post hoc test and are shown as mean ± SEM (*n* = 6). * *p* < 0.05, ** *p* < 0.01. ns: no statistically significant difference.

**Figure 8 ijms-24-12234-f008:**
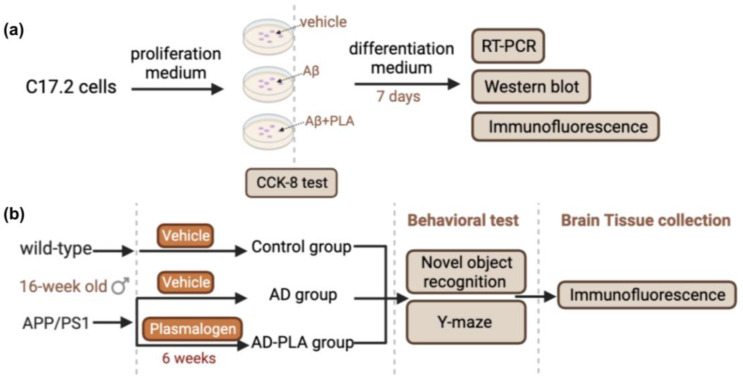
Experimental workflow. (**a**) Schematic illustration of animal experimental process. (**b**) Schematic illustration of cell experimental process.

**Figure 9 ijms-24-12234-f009:**
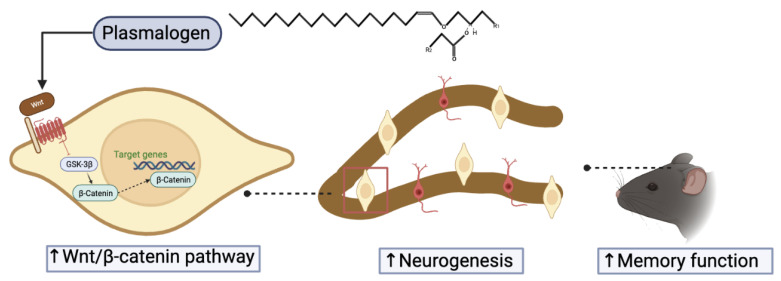
Schematic diagram of PLA-mediated improvement of memory function in AD mice. The combination of plasmalogen and Wnt receptor leads to the inhibition of GSK3β, which increase the level of cytosolic β-catenin. Accumulated β-catenin moves to the nucleus and promotes the transcription of Wnt target genes.

**Table 1 ijms-24-12234-t001:** Percentage of various molecular species in extracted PLA. Cer_AS: ceramide alpha-hydroxy fatty acid-sphingosine; PlasEth: ethanolamine plasmalogens; PlasCholine: choline plasmalogens.

Molecular Species	Percentage (%)
Cer_AS d17:2_18:1	3.22
PlasEtn 18:1_20:5	22.99
PlasEtn 18:1_22:6	28.24
PlasEtn 18:0_22:6	41.8
PlasCholine 16:1_22:6	3.74

## Data Availability

The data used during the current study are available from the corresponding author.

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
