# Peer review of "Plasmalogen Improves Memory Function by Regulating Neurogenesis in a Mouse Model of Alzheimer’s Diseases"

_ijms, 2023, doi:10.3390/ijms241512234_

Round 1

Reviewer 1 Report

My suggestions:

1. Is it possible that mutations in PLA could be potential risk factors for AD or dementia?

2. In the Methods section, I would describe the PLA extraction a little more in detail.

3. Also in the Methos section, I would add a workflow of the experimental process.

4. Was there any change in the inflammatory markers of the mice in the case of PLA treatment? If it was not examined, it would be a nice follow-up study.

5. Did the PLA treatment affect the Tau deposition and neurofibrillary tangle formation?

Reviewer 2 Report

The study „Plasmalogen Improves Memory Function by Regulating Neurogenesis in a Mouse Model of Alzheimer’s Dieases“ by Li et al, analyzed the effect of Plasmalogene (PLA) on neurogenesis in C17.2 cells and mice. 

The manuscript is well written and scientific sound. However i have some points that should be addressed:

Major:

-No ethic statement handling mice experiment is given in the manuscript. If not provided the manuscript cant be published.

-Please show HPLC and LC-MS results of PLA extraction in the supplement and add method to M&M section.

-How was the final concentration of 50mg/ml of PLA extraction was achieved? Which method did the authors used? Bicinchoninic acid assay (BCA)? 

-Did the authors simple used cell lysate or was it adjusted to the same concentration? See BCA.

-What was the rational for the oral intake of 67mg/kg/day - please elaborate.

-Authors should include other effects of plasmalogen on AD into the discussion. See: https://lipidworld.biomedcentral.com/articles/10.1186/s12944-019-1044-1 Plasmalogen reduces gamma-secretase activity and thus reduce Aß and show anti-apoptotic effects. 

Minor:

-Line 102: move sentence in front of figure 2. Guess this will be corrected from the editor.

-Add concentration of PLA and Aß to the figure Legend and Figure (see Fig2,3 and cont.).

These points should be adressed to improve the impact of the manuscript. 

Round 2

Reviewer 1 Report

The authors fulfilled my suggestions. Thank you. 

Author Response

Thank you so much!

Reviewer 2 Report

The authors adressed my comments, however i have some minor points that should be adressed:

Point 1:

-Please add the ethical statement into the manuscripts M&M section.

Point 2:

-In my opinion the results of the molecular species should be shown in the manuscript. Especially since most of them have either DHA or EPA at the sn-2 position. That should be mentioned in the discussion aswell since DHA and EPA shown effects on AD and Aß degradation/aggregation and the alpha secretase. See:

https://www.ncbi.nlm.nih.gov/pmc/articles/PMC6747747/

https://www.ncbi.nlm.nih.gov/pmc/articles/PMC6321163/

https://pubmed.ncbi.nlm.nih.gov/27813426/

https://pubmed.ncbi.nlm.nih.gov/21184792/

The other points are adressed sufficiently.

After adressing the 2 minor points the manuscript is ready to be published.
